# Fresh and Stored Sugar Beet Roots as a Source of Various Types of Mono- and Oligosaccharides

**DOI:** 10.3390/molecules27165125

**Published:** 2022-08-11

**Authors:** Radosław Michał Gruska, Andrzej Baryga, Alina Kunicka-Styczyńska, Stanisław Brzeziński, Justyna Rosicka-Kaczmarek, Karolina Miśkiewicz, Teresa Sumińska

**Affiliations:** 1Department of Sugar Industry and Food Safety Management, Faculty of Biotechnology and Food Science, Lodz University of Technology, ul. Wolczanska 171/173, 90-530 Lodz, Poland; 2Institute of Food Technology and Analysis, Faculty of Biotechnology and Food Science, Lodz University of Technology, ul. B. Stefanowskiego 2/22, 90-537 Lodz, Poland; 3Institute of Agriculture and Food Biotechnology—State Research Institute, ul. Rakowiecka 36, 02-532 Warszawa, Poland

**Keywords:** sugar beet, carbohydrates, oligosaccharides, fructooligosaccharides, extraction, molecular mass

## Abstract

Although sugar beets are primarily treated as a source of sucrose, due to their rich chemical composition, they can also be a source of other carbohydrates, e.g., mono- and oligosaccharides. The study focused on both fresh beet roots and those stored in mounds. Our studies have shown that, in addition to sucrose, sugar beet tissue also comprises other carbohydrates: kestose (3.39%) and galactose (0.65%) and, in smaller amounts, glucose, trehalose and raffinose. The acidic hydrolysis of the watery carbohydrates extracts resulted in obtaining significant amounts of glucose (8.37%) and arabinose (3.11%) as well as xylose and galactose and, in smaller amounts, mannose. An HPSEC liquid chromatography study of the molecular mass profile of the carbohydrate compounds present in the beet roots showed alongside the highest percentage (96.53–97.43%) of sucrose (0.34 kDa) the presence of pectin compounds from the araban group and arabinoxylooligosaccharides (5–9 kDa) with a percentage share of 0.61 to 1.87%. On the basis of our research, beet roots can be considered a potential source of carbohydrates, such as kestose, which is classified as fructooligosaccharide (FOS). The results of this study may be helpful in evaluating sugar beets as a direct source of various carbohydrates, or as a raw material for the biosynthesis of fructooligosaccharides (FOS) or galactooligosaccharides (GOS).

## 1. Introduction

Sugar beet (*Beta vulgaris* L. subsp. *vulgaris*) and sugar cane (*Saccharum officinarum* L.) are the most worldwide utilized plants for sucrose production. The cultivation of sugar cane for the production of sugar dates back to 2000 BC in India, while sugar beet is a relatively new plant and has been cultivated since the end of the 18th century. Currently, around 20% to 25% of world sugar is derived from sugar beet [1,2]. The sugar beet root’s weight can reach up to 2 kg, and the roots usually comprise 75–85% (*w*/*w*) water and up to 21–22% (*w*/*w*) sucrose [3,4,5,6]. Sucrose is the most common, representing more than 98% of the roots’ sugars; however, glucose and fructose are present in smaller amounts [7]. Although sugar beet roots are considered primarily as a raw material for sugar production, they can also be used as a raw material for the direct extraction of other carbohydrates. According to the literature, beet pulp (beet tissue after extraction), besides sucrose, comprises glucose, arabinose, galactose, rhamnose, mannose and xylose [8,9,10,11]. Studies relating to the raw roots of sugar beet are scarce and relate mainly to the quality of sugar beet roots in the light of a raw material for sugar industry [12,13,14,15,16,17,18,19,20]. They concern, in particular, sucrose losses by its inversion into glucose and fructose [14] and the analysis of compounds impairing the efficiency of sucrose production, e.g., pectins [15,16,17], hemicellulose [18,19] or carbohydrates synthesized by contaminating microorganisms (e.g., dextran) [20]. Although researchers mainly focus on pectins and hemicelluloses, they also note the presence of arabinose, galactose, xylose and mannose [21], which can serve as valuable health-promoting compounds or substrates for different chemical or biotechnological processes [22,23,24,25,26,27,28]. However, the direct use of sugar beet roots as a raw material for the production of carbohydrates other than sucrose or direct bioconversion could also be considered. A good example is the case of a fluctuating market situation, with falling sugar prices or falling demand for sugar but a high volume of sugar beet harvest. In such a situation, sugar beet can be considered a direct source of mono- and oligosaccharides, including those of a health-promoting nature, e.g., fructooligosaccharides (FOS). To our knowledge, the literature data concerning this subject are scarce, and sugar beets are treated primarily as a raw material for sucrose production. Due to their abundant yield, sugar beets may serve a good and relatively cheap raw material [1,2], wherein they can be stored for a relatively long period from the time of harvest [13,14]. The analysis of the composition of sugar beets in terms of carbohydrate content with a health-promoting effect seems to complement the existing knowledge about both the chemical composition of sugar beet roots and potential sources of various carbohydrates, including fructooligosaccharides.

Our studies were carried out to determine the carbohydrates content in sugar beet roots directly after harvesting and stored in special mounds (beet clamps) with or without protective covering. The harvesting of sugar beets should be conducted relatively quickly in order to be completed before frost due to the temperate drop in autumn in European countries. Frosts could make it impossible to dig roots out of frozen soil, so the dug beets are usually stored in the form of mounds (beet clamps). The mounds may also be covered to prevent their damage by snow or rainfall. Thus, the method of storage itself may also affect the chemical composition of sugar beet roots, including the carbohydrate profile.

## 2. Results and Discussion

### 2.1. Free Carbohydrates (Mono- and Oligosaccharides) in Sugar Beet Roots

The contents of individual carbohydrates (mono- and oligosaccharides) freshly harvested beet roots and roots stored for three months in covered and uncovered mounds are shown in Table 1, and examples of HPLC chromatograms are shown in Figure 1.

The sugar beet roots contain a number of different carbohydrates, the quantitative composition of which is affected by the way the roots were stored. In all samples tested, sucrose was the predominant free carbohydrate, which was to be expected in a view of the chemical composition of the sugar beet. The amount of sucrose determined was between 16.18 g and 25.18 g/100 g of the sample (Table 1). It can also be noticed that the carbohydrate profile changes depending on the type of root storage. The lowest sucrose content, 16.18 g/100 g, was found in beets stored under cover. The roots of fresh beets comprised about 13.8% more sucrose (18.77 g/100 g), but the highest amount of sucrose was identified in beets stored in uncovered mounds, 25.18 g/100 g, i.e., about 35% more than found in beets stored covered and 25% more than the sample of fresh beets. Other sugars from the group of free carbohydrates in the tested samples were, among others, kestose and galactose, and in much smaller amounts, glucose, trehalose and raffinose. A diversity in the content of these sugars depending on storage method was also observed.

The highest amount of kestose was found in the beet sample stored under cover, i.e., 3.39 g/100 g. For uncovered stored sugar beet, the kestose was 1.24 g/100 g, i.e., 63% lower than for the covered sugar beet sample. The lowest kestose level was in the fresh beet, i.e., 0.30 g/100 g. A smaller difference was found in the galactose contents, which were 0.45, 0.65 and 0.54 g/100 g for beet samples stored under cover, stored uncovered and fresh, respectively. When analyzing the amount of glucose, the trend was similar to ketose content. The highest amount of free glucose was identified in the beet sample stored under cover: 1.25 g/100 g, followed by 0.70 g/100 g in the uncovered sample (almost half as much) and the least, 0.26 g/100 g, in the fresh beets sample. The same trend was also shown for the raffinose content. The highest amount of raffinose was found in the covered sugar beet sample (0.19 g/100 g), then in uncovered stored sugar beet (0.04 g/100 g) and the least in fresh sugar beet (0.01 g/100 g). The trehalose content was 0.02 g/100 g irrespective of the storage method. These results suggest that the carbohydrate profile of sugar beet roots changed during storage and depended on the storage method. It is important to select the appropriate method for sugar beet roots conditioning, make it possible to influence the carbohydrate composition and, consequently, treat them as a source of carbohydrates other than sucrose, e.g., a source of ketose. Some of the results correspond to the results of other authors who have also identified some of the above carbohydrates [21,29]. In addition to sucrose, in fresh roots, the sugars were represented mostly by monosaccharides glucose (26 mg/g) and galactose (54 mg/g), which is consistent with studies by Fares et al. [21] and Rombouts and Thibault [30]. Other sugars were also identified in beets in a region- and climate-dependent amount [21,31,32], e.g., raffinose and fructose at 20 mg/g and 2 mg/g [21], while in our studies, these levels were 0.1 mg/g and 0.1 mg/g, respectively. As mentioned, this may be the result of differences in varieties or climate during cultivation. A very interesting finding was that of kestose in the roots. In the fresh roots, the kestose level was 0.3 g/100 g (3 mg/g) and grew during storage to about 34 g/100 g (340 mg/g). Until now, sugar beet roots have been of interest mainly because of their quality, as a raw material for sugar production. However, our studies show that sugar beet roots may be a potential source of natural fructooligosaccharides (FOS). Fructooligosaccharides (FOS) are constituted by relatively short chains (DP < 20) of β–d-Fru units linked together by (2→1) glycosidic bonds with one terminal Glc residue. Hence, the smallest FOS consist of sucrose to which one, two or three additional Fru units are added to produce 1-kestose [33,34]. Kestose was found mainly in various fruits and vegetables such as banana, garlic, asparagus, leeks, barley, chicory root, artichokes, wheat, onions, legumes as well as honey [35]. For comparison, the kestose content in other plants is garlic (3.3 mg/g), artichoke (1.5 mg/g), shallot (4.5 mg/g) and chicory root (1.7 mg/g) [36]. FOS also can be produced as short-chain FOS (scFOS, a mixture with DP 3–5) from beet sugar (sucrose), exploiting fructosyltransferases and other enzymes from molds, yeasts and a few bacteria [37,38]. Our research broadens the knowledge about the natural sources of kestose by adding sugar beet roots stored in relevant conditions to the potential resources of fructooligosaccharides.

### 2.2. Carbohydrates after Hydrolysis of Complex Carbohydrates (Oligosaccharides) Found in Sugar Beet Roots

Studies of carbohydrates obtained after raw plant material hydrolysis allow us to assess the structure of free complex carbohydrates. Moreover, hydrolysis can be regarded as a preliminary treatment of oligosaccharides for further biotechnological processes. Our research allows us to evaluate the quality of sugar beet roots in terms of the variety of monosaccharides or other carbohydrate resources [22,23,24,25,26,27,28]. Based on the results of the carbohydrate profile of the polysaccharide fraction, after its prior hydrolysis in trifluoroacetic acid (TFA) glucose, arabinose, xylose and galactose were identified as the major compounds (Table 2 and Figure 2). Glucose was detected in the greatest amount and was in the range from 4.55 g to 8.37 g/100 g. The source of glucose in beet samples is mainly the insoluble fiber fraction, i.e., cellulose and its derivatives, a structural component of sugar beet tissue [9,11,21]. The increase in the amount of glucose in the roots (both free glucose and after hydrolysis) may be due to biochemical and microbiological processes that occur during storage [14,38]. Glucose was detected in the greatest amount and was in the range from 4.55 g to 8.37 g/100 g. The source of glucose in beet samples is mainly the insoluble fiber fraction, i.e., cellulose and its derivatives, a structural component of sugar beet tissue [9,11,21].

According to the sugars identified, arabinose was the second, followed by xylose and galactose. This is because these form the skeleton of pectin compounds.

Beet pectins consist mainly of arabans, also called arabinans with the main chain made of arabinose [21,35]. Xylose and small amounts of galactose are part of the pectin side chains [21]. There are differences in the level of monosaccharides after hydrolysis in fresh and stored beets. The beets stored for three months under the cover expressed the highest levels of glucose, arabinose, xylose and galactose compared to the sample of fresh and uncovered ones. The amount of glucose in covered beets was, on average, two times higher (8.37 g/100 g) than that in fresh and uncovered beet roots, 4.95 g and 4.55 g/100 g of sample, respectively. Significantly smaller differences in the amount of arabinose were observed: 3.11 g, 2.81 g and 2.96 g/100 g for the covered stored beets, fresh beets and uncovered stored beets, respectively. Xylose and galactose showed a very similar trend. On this basis, it can be assumed that the higher proportion of polysaccharide compounds of cellulose and pectin (mainly arabanas) was noted in sugar beet roots covered during storage. The significant differentiation in the carbohydrate profile in sugar beet tissue was presumably due to the composition of pectin compounds, mainly arabans and cell wall tissue residues, i.e., cellulose, which was shown by our findings and the research of other authors [21].

### 2.3. Study of the Molecular Mass Profile of Carbohydrate Compounds Present in Sugar Beet Roots

The analysis of molecular masses allows us to assess the size of individual carbohydrates that have been extracted from the roots. This is particularly helpful in the analysis of complex carbohydrates (oligosaccharides), which can be polymers of various sizes (e.g., dextran, cellulose or hemicellulose). Knowledge of their molecular mass and percentage share makes it possible to assess how large molecules and in what quantity they were extracted under experimental conditions. This can be helpful in establishing extraction conditions when considering roots as a source of mono- and oligosaccharides. The results of molecular weight profile analysis of the beet roots’ carbohydrate compounds after complex polysaccharides hydrolysis (Table 3) indicate a strong correlation with the profile of the free carbohydrates and carbohydrates after hydrolysis. The highest percentage (on average from 96.53% to 97.43%) among the indicated molecular masses in all the samples is 0.34 kDa, attributed to the molecular mass of sucrose. It is related to the amount of sucrose determined as free carbohydrates in the free sugar profile in the sugar beet root tested. The next highest weight fraction was the mass between 5.31 kDa and 9.28 kDa, which is attributed to the presence of pectic compounds from the araban group and arabinoxylooligosaccharides. Its percentage in the studied samples ranged from 0.61 to 1.87% and simultaneously correlates with the carbohydrate profile after acid hydrolysis of the analyzed samples.

The difference in the quantity of the molecular mass compounds in the range 5.31–9.28 kDa depending on the beet freshness was noted. It was only 0.61% in fresh beets, 1.69% in covered storage beets and 1.87% in uncovered storage beets. Compounds with an average weight of 1.00–1.83 kDa, representing 0.3–0.9% of isolated carbohydrates and characteristic of arabinooligosaccharides or xylooligosaccharides, were also identified. In the sugar beet, compounds with molecular weight between 44.14 kDa and 109.38 kDa were present in the lowest percentages (0.08–0.12%). These molecular weight values are characteristic for cellulose, hemicellulose and derivatives [39].

## 3. Materials and Methods

### 3.1. Sugar Beet Roots

Roots of sugar beet (*Beta vulgaris* L. subsp. *vulgaris*) classic variety were originated from a plantation of a Polish sugar factory located in the center of the country. Beet roots were harvested in September in 2019. The roots after harvesting were divided into three parts and used for the research:Fresh roots directly from the field;Roots stored in the mounds (beet clamps) for three months (from September to December). The hills were not covered;Roots stored in the mounds (beet clamps) for three months (from September to December). To prevent rain, the mounds were covered with an airy and waterproof fabric.

During storage in mounds (uncovered and undiscovered), the following weather conditions prevailed (Table 4).

In order to obtain representative analytical results, approximately 50 kg of roots were taken at random, milled and blended to ensure the homogeneity of the sample.

### 3.2. Determination of Free Carbohydrates in Sugar Beet Roots

A qualitative and quantitative analysis of soluble carbohydrates was performed using high-performance liquid chromatography [40,41]. First, 2 g ± 0.1 g representative samples of milled and blended sugar beet roots were mixed with 10 cm^3^ of HPLC-grade redistilled water and incubated in a water bath at a temperature of 85 °C for 1 h with shaking. Afterwards, the samples were cooled and centrifuged at a temperature of 20 °C for 15 min at 4600 rpm. Then, 2 cm^3^ of supernatant was collected and filtered into HPLC measuring vials by nylon syringe filters with a pore size of 0.2 μm. HPLC-RI was used for analysis: UHPLC+ Dionex UltiMate 3000 system (Thermo Fisher Scientific Inc., Waltham, MA, USA) equipped with a refractive index detector (Shimadzu, Kioto, Japan). The following conditions were applied: Rezex column RPM Monosaccharide Pb^2+^ New Column, 8 μm, 7.8 × 300 mm; Eluent: 100% HPLC purity water; isocratic gradient; column preparation: 100% eluent in 20 min; analysis: 100% eluent in 35 min; flow rate: 0.6 mL/min; column temperature: 80 °C; injection volume: 0.01 cm^3^ (10 μL); detection: Shodex R1–101, temperature 40 °C.

### 3.3. Determination of Sugar Beet Roots Carbohydrates after Hydrolysis

The purpose was to extract the carbohydrates from the roots and then hydrolyze them to monomers. The hydrolysis of sugar beet carbohydrates was performed as described in the literature [40,41]. Firstly, 10 mg ± 0.1 mg representative samples of milled and blended sugar beet roots were mixed with 4 mL of 2 M trifluoroacetic acid (TFA) and incubated at a temperature of 100 °C for 150 min. After hydrolysis, the samples were immediately cooled and centrifuged at a temperature of 20 °C for 15 min at 4600 rpm. Then, 2 cm^3^ of the supernatant was introduced into 5 cm^3^ tubes and dried in a stream of nitrogen for TFA evaporation. The dry precipitate was dissolved in 1 mL of HPLC-grade water, and the solution was filtered into HPLC measuring vials through nylon syringe filters with a pore size of 0.2 µm and subjected to HPLC-RI analysis, as described in Section 3.2.

### 3.4. Determination of the Molecular Mass of Carbohydrates

The molecular mass distribution profile was determined by HPSEC liquid chromatography [42]. A milled and blended sugar beet root representative sample of 120 mg ± 0.1 mg was mixed with 20 cm^3^ of dimethyl sulfoxide (DMSO) and shaken for 3 h at room temperature. Then, the samples were left for 24 h at room temperature without agitation. The samples were incubated at a temperature of 90 °C in a water bath for 1 h and cooled. Next, 0.1 cm^3^ of the prepared solution was transferred to chromatographic vials and mixed with 0.9 cm^3^ of HPLC grade water. Dextran molecular standards 1, 5, 25, 80, 670 and 1300 kDa were used as reference. The following conditions were used for the chromatographic analysis: column: PolySep-GFC-P 1000 LC, 300 × 7.8 mm; column temperature: 30 °C; eluent: 100% HPLC-grade water; flow rate: 0.4 cm^3^/min; analysis time: 40 min; injection volume: 0.01 cm^3^ (10 μL); detector RI: 40 °C.

### 3.5. Statistical Analysis

All experiments were carried out in triplicate. The results were expressed as mean ± standard deviation (SD). Statistical analyses were performed using Statistica 13.1 software (StatSoft, Cracow, Poland).

## 4. Conclusions

Sugar beet roots may serve as a raw material for biotechnological use, rich in other carbohydrates apart from sucrose, including both mono- and oligosaccharides. Insignificant amounts glucose (0.26% *w*/*w*), galactose (0.54% *w*/*w*) or kestose (0.30% *w*/*w*) were found in freshly harvested roots. Moreover, the quantitative composition of carbohydrates varied depending on the method of beet root storage with glucose and raffinose increase (respectively, by 0.99% *w*/*w* and 0.18% *w*/*w*, in covered roots) as well as galactose (increase by 0.11% *w*/*w* in uncovered ones). Interestingly, the fructooligosaccharide kestose level rose more than 11 times (up to 3.39% *w*/*w* and equal to 33.9 mg/g) in the roots stored under cover compared to the fresh ones. After the hydrolysis of free carbohydrates in the tested beet samples, in addition to the disaccharide sucrose, macromolecular compounds were found that can be attributed to arabans, arabinoxylooligosaccharides and xylo- and arabinooligosaccharides in an amount of approximately 0.6% to 1.9%. These studies can be used for the sugar beets’ evaluation as a direct source of various types of mono- and oligosaccharides, e.g., kestose, or a raw material for other types of carbohydrates after hydrolysis or bioconversion. Carbohydrates can be sourced from both fresh and stored sugar beet roots, but the storage conditions of the roots strongly determine the profile and amount of carbohydrates, in particular, the kestose content.

## Figures and Tables

**Figure 1 molecules-27-05125-f001:**
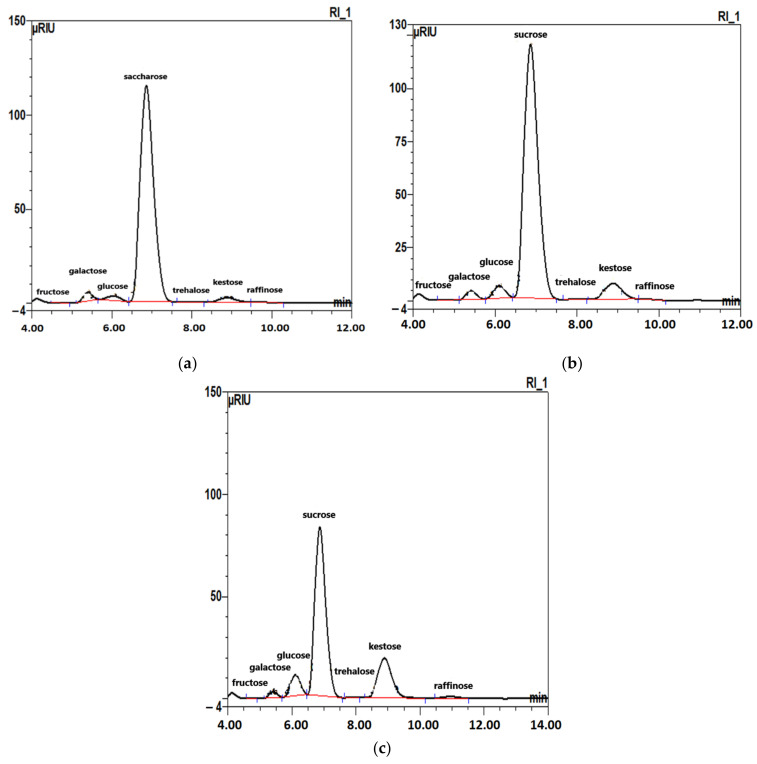
HPLC chromatograms of free carbohydrates in sugar beet roots: (**a**) fresh roots; (**b**) roots stored in uncovered mounds (beet clamps) for three months; (**c**) roots stored for three months in covered (rain and snow-proof) mounds.

**Figure 2 molecules-27-05125-f002:**
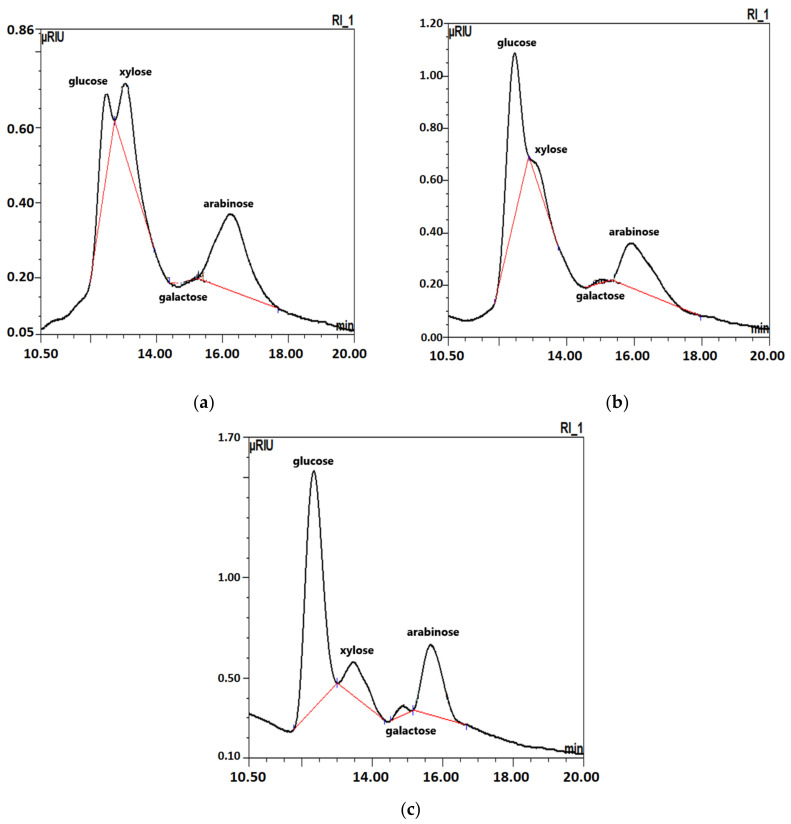
HPLC chromatograms of the profile of free sugars after the hydrolysis of complex carbohydrates of sugar beet roots: (**a**) fresh roots; (**b**) roots stored in uncovered mounds (beet clamps) for three months; (**c**) roots stored for three months in covered (rain and snow-proof) mounds.

**Table 1 molecules-27-05125-t001:** The content of free carbohydrates in sugar beet roots (% *w*/*w*).

Carbohydrates	Fresh Roots	Roots Stored for Three Months in	
		Uncovered Mounds	Covered Mounds
Fructose	0.01 ± 0.00	0.01 ± 0.00	ND
Glucose	0.26 ± 0.01	0.70 ± 0.01	1.25 ± 0.02
Galactose	0.54 ± 0.01	0.65 ± 0.01	0.45 ± 0.01
Saccharose	18.77 ± 0.18	25.18 ± 0.24	16.18 ± 0.20
Trehalose	0.02 ± 0.00	0.02 ± 0.00	0.02 ± 0.00
Kestose	0.30 ± 0.01	1.24 ± 0.01	3.39 ± 0.03
Raffinose	0.01 ± 0.00	0.04 ± 0.00	0.19 ± 0.01

ND—not detected.

**Table 2 molecules-27-05125-t002:** The content of carbohydrates after the hydrolysis of complex carbohydrates in sugar beet roots (% *w*/*w*).

Carbohydrates	Fresh Roots	Roots Stored for Three Months in	
		Uncovered Mounds	Covered Mounds
Glucose	4.95 ± 0.05	4.55 ± 0.04	8.37 ± 0.11
Xylose	0.90 ± 0.01	0.33 ± 0.02	1.15 ± 0.02
Galactose	0.06 ± 0.00	0.05 ± 0.01	0.20 ± 0.01
Arabinose	2.81 ± 0.03	2.96 ± 0.03	3.11 ± 0.03

**Table 3 molecules-27-05125-t003:** Molecular masses of carbohydrates after hydrolysis of complex carbohydrates in sugar beet roots.

Sample	Mass (kDa)	Percentage (%)
Fresh sugar beet roots	89.21 ± 0.80 9.28 ± 0.10 1.76 ± 0.02 1.40 ± 0.02 1.03 ± 0.01 0.34 ± 0.01	0.10 ± 0.01 0.61 ± 0.01 0.92 ± 0.02 0.76 ± 0.01 0.88 ± 0.01 96.73 ± 0.71
Roots stored for three months (uncovered mounds)	109.38 ± 0.89 5.31 ± 0.03 1.43 ± 0.02 1.00 ± 0.01 0.34 ± 0.01	0.12 ± 0.01 1.87 ± 0.02 0.65 ± 0.01 0.83 ± 0.01 96.53 ± 0.70
Roots stored for three months (covered mounds)	44.14 ± 0.35 5.59 ± 0.02 1.83 ± 0.01 1.54 ± 0.01 0.34 ± 0.01	0.08 ± 0.01 1.69 ± 0.01 0.44 ± 0.01 0.36 ± 0.01 97.43 ± 0.60

**Table 4 molecules-27-05125-t004:** Weather conditions during the storage of sugar beet roots.

Month	Temperature (°C)			Total Rainfall (mm)	Insolation (h)
	Average	Minimum	Maximum		
September	15	3	26	60–70	160–180
October	10	−4	21	40–55	120–140
November	6	−3	14	<20	40–50

## Data Availability

Data available on request due to restrictions, e.g., privacy or ethical The data presented in this study are available on request from the corresponding author. The data are not publicly available because they are part of the authors’ own research.

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
