# Peer review of "Fresh and Stored Sugar Beet Roots as a Source of Various Types of Mono- and Oligosaccharides"

_molecules, 2022, doi:10.3390/molecules27165125_

Round 1

Reviewer 1 Report

The research is interesting in the area of the analysis of beet roots as a source of various types of mono and oligosaccharides. Nevertheless, the manuscript needs to be improved in order to provide enough information to justify its  importance:

    Lines 25-26. Please rephrase the sentence

· The introduction part should better explain the originality of this paper, which results in a little lacking in this version.

   There is  a lack of a description of statistical analysis (the software used, if any)

·      Lines 270-272. It should be moved to the Results and discussions

Please conclude according to the main objectives of the research and complete with more practical applications of the authors’ findings.

Author Response

Response to Reviewer 1 Comments

Dr Radosław Gruska

Manuscript ID: Molecules-1842286

Type of manuscript: Full Length Article

Title: Fresh and stored sugar beet roots as a source of various types of mono and oligosaccharides

Dear Reviewer (1),

We would like to thank the Reviewer for the valuable and stimulating comments on our manuscript. These comments and suggestions helped us to prepare its revised version. All the changes addressed according to the Reviewer’s suggestions have been highlighted in red.

Point 1: Lines 25-26. Please rephrase the sentence

 Response 1: We agree with the Reviewer that the “The other carbohydrates may be substrates for the biosynthesis of fructooligosaccharides (FOS) or galactooligosaccharides (GOS)” sentence was awkward.

It has been converted to:

The results of this study may be helpful in evaluating sugar beets as a direct source of various carbohydrates, or as a raw material for the biosynthesis of fructooligosaccharides (FOS) or galactooligosaccharides (GOS).

Point 2: The introduction part should better explain the originality of this paper, which results in a little lacking in this version.

Response 2: The following text was added in the introduction:

In such a situation, sugar beet can be considered as a direct source of mono- and oligosaccharides, including those of a health-promoting nature, e.g. fructooligosaccharides (FOS). To our knowledge, the literature data concerning this subject is scarce, and sugar beets are treated primarily as a raw material for sucrose production. Due to their abundant yield, sugar beets may serve a good and relatively cheap raw material [1,2], wherein they can be stored for a relatively long period from the time of harvest [13,14]. The analysis of the composition of sugar beets in terms of carbohydrate content with a health-promoting effect seems to complement the existing knowledge about both the chemical composition of sugar beet roots and potential sources of various carbohydrates, including fructooligosaccharides.

Point 3: There is a lack of a description of statistical analysis (the software used, if any).

Response 3: In section 4.5. Statistical analysis the information concerning statistical analysis was added:

All experiments were carried out in triplicate. The results were expressed as a mean ± standard deviation (SD). Statistical analysis was performed using Statistica 13.1 software (StatSoft Poland).

Point 4: Lines 270-272. It should be moved to the Results and discussions.

Response 4: We agree with the Reviewer and these sentences have been moved to Results and Discussion (lines 124 – 125). The numbering of the bibliography has been changed.

Point 5: Please conclude according to the main objectives of the research and complete with more practical applications of the authors’ findings.

Response 5: These studies can be used for the sugar beets evaluation as a direct source of various types of mono- and oligosaccharides e.g. kestose, or a raw material for other types of carbohydrates after hydrolysis or bioconversion. Carbohydrates can be sourced from both fresh and stored sugar beet roots, but the storage conditions of the roots strongly determine the profile and amount of carbohydrates, in particular the kestose content.

Yours sincerely,

Radosław Gruska

Reviewer 2 Report

This article describes a new study on the sugar content of sugar beet, one of today's main sources of sucrose.
The work carried out has made it possible to isolate minority sugars from this raw material, including mainly kestose and galactose. In addition, hydrolysis of the extracts has also yielded significant amounts of glucose, arabinose, xylose, galactose and mannose.
This has made it possible to establish that sugar beet can be a source for obtaining mono- and oligosaccharides of chemical and biological interest. This is a remarkable contribution.
The work is robust and the article is well written and organised.
Consequently, it is my opinion that it should be accepted for publication in the journal Molecules, without modification.

Author Response

Response to Reviewer 2 Comments

Dr Radosław Gruska

Manuscript ID: Molecules-1842286

Type of manuscript: Full Length Article

Title: Fresh and stored sugar beet roots as a source of various types of mono and oligosaccharides

Dear Reviewer (2),

Thank you very much for your valuable review of our manuscript. We are pleased that our research attracted the attention of the Reviewer and was recommended for publication.

Yours sincerely,

Radosław Gruska
